# Role Identity, Dissonance, and Distress among Paramedics

**DOI:** 10.3390/ijerph19042115

**Published:** 2022-02-13

**Authors:** Justin Mausz, Elizabeth Anne Donnelly, Sandra Moll, Sheila Harms, Meghan McConnell

**Affiliations:** 1Peel Regional Paramedic Services, Brampton, ON L6V 4R5, Canada; 2Department of Health Research Methods, Evidence and Impact, McMaster University, Hamilton, ON L8S 4L8, Canada; 3School of Social Work, The University of Windsor, Windsor, ON N9A 0C5, Canada; donnelly@uwindsor.ca; 4School of Rehabilitation Sciences, McMaster University, Hamilton, ON L8S 1C7, Canada; molls@mcmaster.ca; 5Department of Psychiatry and Behavioural Neurosciences, McMaster University, Hamilton, ON L8N 3K7, Canada; harmssh@hhsc.ca; 6Department of Innovation in Medical Education, Faculty of Medicine, The University of Ottawa, Ottawa, ON K1G 5Z3, Canada; meghan.mcconnell@uottawa.ca

**Keywords:** public safety personnel, first responders, mental disorders, mental health, wellbeing, trauma, operational stress injuries, post-traumatic stress injuries, role identity theory, qualitative research

## Abstract

Role identity theory describes the purpose and meaning in life that comes, in part, from occupying social roles. While robustly linked to health and wellbeing, this may become unideal when an individual is unable to fulfill the perceived requirements of an especially salient role in the manner that they believe they should. Amid high rates of mental illness among public safety personnel, we interviewed a purposely selected sample of 21 paramedics from a single service in Ontario, Canada, to explore incongruence between an espoused and able-to-enact paramedic role identity. Situated in an interpretivist epistemology and using successive rounds of thematic analysis, we developed a framework for role identity dissonance wherein chronic, identity-relevant disruptive events cause emotional and psychological distress. While some participants were able to recalibrate their sense of self and understanding of the role, for others, this dissonance was irreconcilable, contributing to disability and lost time from work. In addition to contributing a novel perspective on paramedic mental health and wellbeing, our work also offers a modest contribution to the theory in using the paramedic context as an example to consider identity disruption through chronic workplace stress.

## 1. Introduction

“You’re at your peak here (having just attended a call for a cardiac arrest), and you feel good about yourself. And then some guy calls for ‘I can’t sleep’. I actually felt like a paramedic, (and) now I’m your taxi driver who’s taking you to the hospital because you can’t sleep.”(‘Shawn’)

Shawn is an experienced advanced care paramedic who I interviewed to ask about how role identity—how we see ourselves in relation to the roles that we hold in society [1]—intersects with his mental health and wellbeing as a paramedic. The question is both important and timely, given the growing recognition of the mental health challenges that paramedics face because of their work. In a recent cross-sectional survey of public safety personnel in Canada, one in four participating paramedics met the screening criteria for post-traumatic stress disorder (PTSD), one in three for depression, and one in three for an anxiety disorder [2]. All told, nearly half of the surveyed paramedics met the screening criteria for at least one mental disorder [2], which—combined with high rates of chronic pain [3], exposure to trauma [4], substance use [2], and a history of childhood abuse [5]—conspired to place the cohort at an alarmingly increased risk of suicide [6].

Although several studies have attempted to define and quantify the impact of various stressors on the risk of mental illness among paramedics, a precise answer has been elusive. Much of the extant research has relied on cross-sectional surveys that view the topic of paramedic mental health through a biomedical (or diagnostic) lens and have tended to be atheoretical. Although informative, this approach risks missing part of the picture. Theoretically informed social science perspectives have contributed much to our understanding of health and wellbeing [7] and have the potential to shed new light on the topic among paramedics. One early line of inquiry that shows promise is the use of role identity theory [8] to study paramedic mental health.

### 1.1. Role Identity Theory

Originating within the broader family of identity theories [1] and flowing from the tenets of symbolic interactionism [9], role identity theory explains that individuals experientially construct a sense of self through the enactment of social roles [10]. Roles are relational positions within society to which there are attendant behavioral expectations and norms, including various attitudes, values, and beliefs [11]. Being a parent, for example, carries with it responsibilities that are articulated in law, ethics, and social scripts [12]. Where a role holds particular salience—defined as the perceived importance [10]—for the individual, the role becomes a central part of the person’s sense of self, providing an answer to the existential question “who am I?” Role identities serve as a signal to ourselves and to others about how we fit into society, and the performative nature of role identities means that perceptions of self hinge to a degree on how we think others see us [13]. Role-affirming experiences help reinforce the stability of the identity [14], and the resulting senses of purpose and meaning that come from competently fulfilling an important role have been robustly linked to physical, emotional, and psychological wellbeing [15]. This has been observed, for example, among older volunteers, who, after retirement, derive meaning through community service [10,16,17].

### 1.2. Paramedic Role Identity 

A four-dimension paramedic role identity has been previously defined [18]. Within this construction, personal and professional fulfillment can be drawn from helping people in need (caregiving); finding excitement in the dramatic aspects of paramedic work (thrill seeking); deriving a sense of self-efficacy from competently performing challenging work (capacity); or the altruism of providing an important community service (duty) [18]. Viewing paramedic mental health through a role identity lens may provide useful insights. Due to the fact that paramedics hold a respected position in society, role identity is inherently high stakes, both in its attendant function of responding to life-threatening emergencies, and in its social capital, where paramedics are often portrayed in a heroic light [19]. If the paramedic role identity holds particular salience for an individual, it follows that their sense of purpose and meaning in life may run equally deep. Where this may become problematic, however, is if the individual is unable to fulfill the attributes of the role in the manner that they feel (or believe others feel) that they should—what Thotis and others have called “identity-relevant disruptive” events [20,21]. Although usually discussed in the context of discrete life events such as the loss of a job or the death of a spouse [22], it may be reasonable to extrapolate the concept to more nebulous chronic stressors. Shawn’s comments at the beginning of the paper are illustrative of a discrepancy between perceived and enacted role identity that, over the course of a career, could result in potentially significant chronic stress. We sought to explore this issue of incongruence and its implications for mental health among paramedics by asking the question “what happens when a paramedic is unable to fulfill the attributes of the role in the manner they believe they should?”

## 2. Materials and Methods

### 2.1. Overview

We positioned this study within an interpretivist epistemology [23], under the methodological banner of generic approaches to qualitative research [24], and adopted what Varpio and colleagues [25] call a “fully theoretically informed” study design. Sensitized by an existing definition for paramedic role identity [18], we conducted in-depth, multistage, semistructured interviews with a purposely selected sample of 21 paramedics from a single, large, urban paramedic service in Ontario, Canada. Our goal in this study was to explore the incongruence between an espoused and able-to-enact role identity and its potential implications for mental health and wellbeing. In practical terms, this meant using “if-then” propositions of the theory to answer five specific questions:How do the dimensions of paramedic role identity align with the ways in which our participants see themselves in relation to their role?Relatedly, is anything “missing” in terms of new dimensions of paramedic role identity?How does incongruence between an espoused and able-to-enact role identity (what we call “role identity dissonance”) manifest?What consequences result from role identity dissonance in terms of emotional or psychological distress?Finally, how is role identity dissonance reconciled?

### 2.2. Theoretical Orientations and Approach

Due to the fact that our construct of interest flows broadly from the tenets of symbolic interactionism [26] and role relationships themselves are necessarily performative [1], we acknowledge the inherent centering of subjective experience in making sense of the lived world. This aligns with interpretivist thinking [23] in understanding that reality is experienced subjectively and negotiated collaboratively, allowing us to explore the richness of multiple, and at times, seemingly divergent or contradictory truths. In interpretivist approaches to research, the role of the investigator in the co-construction of knowledge is embraced rather than bracketed out. Strategies to distance the influence of the researcher from the topic of study or make the research process “objective” tend to be eschewed in favor of being transparent about the positionality and role of the researcher in the construction of knowledge. We discuss this in more detail in our section on reflexivity.

### 2.3. Ethics and Consent

Ethics review for this study was provided by the Hamilton Integrated Research Ethics Board (project number 5599) and approval for the research within the study site was provided jointly by the paramedic leadership team and the elected local executive committee of the union representing the paramedics. All participants provided informed, written consent to participate, were assured of confidentiality, and had the option to withdraw from the study at any point. We also put in place additional safeguards for the participants given the sensitivity of the interview subject. This included taking breaks when discussing difficult topics, debriefing participants following each interview, providing each interviewee with a list of mental health resources, and having referral and crisis procedures for distressed participants (a contingency plan that, fortunately, was not required).

### 2.4. Setting and Context

Our study took place pre-COVID-19 in a single paramedic service in Ontario, Canada. The publicly funded, lower-tier municipal service employs more than 700 primary and advanced care paramedics who respond to an average of 130,000 emergency calls per year, making the service the second largest in the province by staffing and caseload. The topic of mental health has been particularly salient within the service in recent years after the deaths by suicide of two senior paramedics. 

### 2.5. Researcher Characteristics and Reflexivity

This investigation forms one part of my doctoral dissertation in health research methods. For clarity, where I write in the first person singular, I am referring to actions, procedures, decisions, or interpretations that I (JM) make as the first author and lead investigator on the study. Where we write in the first-person plural, we are referring to actions, procedures, or decisions that the research team made as a group. Our team blends a variety of disciplinary backgrounds, including social work, occupational therapy, psychiatry, cognitive psychology, and paramedicine, with each member having experience in a variety of research approaches, including qualitative methodologies. I am also a practicing paramedic at the study site, thus positioning me as an insider [26] in the community. This afforded me access and insight into the phenomenon under study that would otherwise be difficult to obtain, but it came, however, with the acknowledged risk of unchallenged assumptions shared between me and the participants. On balance, we felt that the affordances outweighed the risks; my relationship with the participants as a respected colleague provided me a unique position as a trusted confidant. Since I have shared many of the same experiences as my interview participants, it provided me the legitimacy to ask difficult questions and a common language to interpret responses. To help counterbalance my insider perspective, I engaged in a variety of reflexive processes, including reflective journaling and memoing common to qualitative methods [27], debriefing interviews with our research assistant, and bringing annotated interview excerpts to the members of the research team for discussion. The goal here was not to eliminate or even substantially reduce “bias” per se but to instead build transparency in the research process and incorporate different perspectives into the interpretation of findings in a manner that is still in alignment with our guiding conceptual framework. 

### 2.6. Recruitment and Sampling

Our sampling strategy not only followed the methodological principles of purposive sampling as described by Charmaz [28], recruiting a sample of 21 paramedics using maximum variation across demographic characteristics, but also made an effort to saturate various conceptual categories relevant to our research questions. This included recruiting participants who had taken an occupational stress leave (*N* = 7) or who had been diagnosed with or were receiving treatment for a work-related mental health problem (*N* = 6). We solicited participation through workplace and union email list servers, closed workplace social media groups, and in person during the fall 2019/winter 2020 Continuing Medical Education (CME) sessions. All recruitment was handled by the principal investigator—who is himself a practicing paramedic in the study site—and involved an explanation of the study goals and the range of participant experiences the team was interested in exploring (i.e., variety in gender, age, years of experience, level of clinical certification, employment classification, and lived experience with work-related mental illness). 

### 2.7. Data Collection

We used multistage, semistructured interviews for data collection. I interviewed every participant at least once, with most participants providing two interviews and two providing three. The interviews loosely followed a biographical narrative approach [29]. Each interview referenced a semistructured interview guide developed through consensus among the research team, with sensitizing concepts drawn from a review of the paramedic mental health literature and our conceptual framework [7]. As an indicator of role salience [10], I asked each participant the following question: “Imagine your sense of self as a pie chart. How big a ‘slice’ does being a paramedic get and why?”

Since our theoretical framework posits that identity-challenging events may be more distressing for particularly salient role identities, I was particularly interested in the effects of incongruence between espoused and able-to-enact role identity among participants for whom the identity formed a central part of their sense of self (i.e., “it [paramedic identity] gets all of my pies” ‘Johnathan’). 

I conducted the interviews in person at workplace facilities, taking care to ensure the participant’s privacy. I recorded the conversation, made handwritten notes to document initial observations and flag ideas for follow-up, and then wrote (or typed) more detailed notes after the interviews. Each interview lasted between 60 and 120 min, and we provided participants with a gift card for a vendor of their choosing in the amount of CAD 90.

### 2.8. Data Analysis

We used a web-based transcription program (Temi, San Francisco, CA, USA) to generate transcripts from audio recordings. My assistant and I reviewed each completed transcript to correct for errors, add in paralanguage (long pauses, laughing, crying, etc.), and edit out nonrelevant false starts or fillers to achieve a “clean verbatim” level of transcription. This review also served as a useful first level of analysis in becoming oriented to the data. Finalized transcripts were prepared as Microsoft Word documents and imported into NVivo (QSR International, Doncaster, Australia) for manual coding. 

I used successive rounds of open [30] and focused [31] coding to answer our research questions. First, drawing on the extant dimensions of paramedic role identity as sensitizing concepts, I looked for alignment in the ways in which the participants spoke about their motivations to perform work and how they derive meaning from it but being cognizant not to “force” data into rigid a priori categories. Second, I looked for ways in which the participants spoke about their work that did not align with the existing dimensions of paramedic role identity. I used descriptive in vivo codes composed of the participants’ own words to “flag” these areas of divergence. In subsequent rounds of focused coding, I grouped “divergent” perspectives by conceptual similarity. I assigned each (N = 2) with a new label using gerunds (protecting and problem solving) and then compared each with the definitions of the extant dimensions as articulated in the original publication to ensure that they were distinct constructs. 

Lastly, I reviewed the transcripts for areas of incongruence between (what I interpreted as) espoused dimensions of role identity and the participants’ ability to fulfill the attributes of that role. I mostly inferred this incongruence from the more readily apparent dissonance articulated by the participants, often cited as a source of workplace stress. I organized codes thematically, using gerunds in successive rounds of open and focused coding to define the processes of how role identity dissonance manifests and may or may not be reconciled. 

Returning here to the issue of role salience, I drew on (and quote more heavily from) participants for whom paramedic identity formed a large part of their sense of self. This was in part a deliberate methodological choice and also a natural happenstance explainable by the theory. Role identities are arranged hierarchically within the self [32], and because the effects of identity-disruptive events are felt more keenly for more salient role identities [15], the consequences of incongruence were more pronounced, making the phenomenon more accessible analytically speaking. That is not to say, however, that any participants were excluded from the analysis; it is only that, in presenting the findings, there were more participants of some than others.

### 2.9. Note to Readers

Our interviews touched on difficult topics, and the participants occasionally used strong language, which, for the purposes of presenting their experiences faithfully, we have not edited or redacted. The interview excerpts we present may be triggering to readers who have personal or professional experience with life-threatening illness or injury, violence (including child abuse and intimate partner violence), mental illness, or suicide. Please read carefully. 

## 3. Results

### 3.1. Participant Characteristics

Our sample of 21 participants included women (*N* = 11) and men (*N* = 10) between 27 and 56 (mean 36 ± 6.7) years of age who had between 1 and 28 (mean 12, ±6.5) years of experience as paramedics and worked in a front line, supervisory, or special operations role at both the primary and advanced care provider classifications. We also recruited participants who had previously been or were currently members of the service’s peer support team—a group of trained volunteers who provide empathic support to colleagues. Finally, our sample also included participants who were in the process of returning to work after an occupational stress leave. All names presented are gender-consistent pseudonyms.

### 3.2. Question 1: Alignment with the Existing Dimensions 

The participants generally used language that was quite consistent with the definitions provided by Donnelly and colleagues (see Table 1 for examples). Of the four dimensions, caregiving, thrill-seeking, and capacity tended to come through the strongest and generally in that order. The thrill-seeking dimension presented an unexpected division among the participants: most of the paramedics I spoke with who aligned with the dimension made veiled, almost self-conscious references to enjoying the excitement of emergency work. 

“I don’t want to say it’s an adrenaline junkie thing, but at the same time, like, what am I learning (by being in a less busy part of the city) … Like it’s not bad. But, I was like, I want to do other calls.”(Rowan)

Conversely, other participants were very explicit about seeking out the ‘rush’ of paramedic work.

“We’re all ‘adrenaline junkies’, we do it for the adrenaline rush. You can say you do it for the patients, or this or that, (but) no, we do it because (we) want to drive lights and sirens and (we) want to be put on the spot to make a decision that makes the difference between life and death. You know it, I know it, everybody knows it.”(Shawn)

### 3.3. Question 2: New Dimensions: Problem Solving and Protecting

We defined the features of two new dimensions of paramedic role identity. The first involves a curiosity-driven and scientifically informed desire to solve problems. Paramedic training includes several courses in anatomy, physiology, pathophysiology, and biology that the participants described as giving the work a “mental” quality: “I’ve always had an interest in the science aspect” (Dean). Extending their natural curiosity to paramedic work provided a deep sense of fulfilment in drawing on their science training to solve clinical puzzles.

“Another (patient) was having a stroke. It was a DVT (deep vein thrombosis) that became a stroke, and then I started thinking ‘wait a minute, shouldn’t the clot be in your lungs?’ So, I brought it up with the doctor and they’re like ‘oh, we’ll take a look’ and sure enough there was a hole through the septum (in her heart) where the clot went into the other side (of her heart) and then into her brain. … But, like, the fact that I came up with that independently made me feel very good about myself.” (Seamas)

The second dimension refers to a desire among participants to not only help people in need (caregiving) but to actively protect people from harm: “(We are here) to be that safety net when everything else fails” (David). Although not exclusively, this sense of protecting others came through particularly strongly when the participants spoke about calls involving children or other vulnerable groups.

“(I had) this nasty patient, some alcoholic old man who was trying to justify to me why it was okay for him to beat his wife because she did not want to have sex with him the night before, and I said ‘Look at me: shut the fuck up. I don’t want to hear it.’”(Elaine)

The difference between caregiving and protecting is perhaps subtle, but it carried a great deal of importance for the participants and appeared to hinge on the distinction between reacting to problems versus proactively preventing harm. This desire to have proactive “upstream” impacts on people’s lives was what attracted many of the participants to the profession. The cruel irony is that the majority of paramedic work is inherently reactive—responding to emergencies after they have occurred and being left to pick up the pieces. Invariably, however, the participants would observe other problems in the making at the scenes they attended and want to intervene. For example, Elaine later described a call she attended for a woman who had overdosed and was unconscious with two young children at home.

“You start to get that protective instinct, and we had to just leave them (the children) there, and they got sent to their aunt’s, I think, that night. But then they were just going right back … Seeing kids who are being looked after by CAS (the Children’s Aid Society) and they’re just dumped back into a house that’s awful in every single possible way because there’s nothing else they can do. I just thought we would be more a part of the solution and it just seems like we’re a smokescreen.”

The incongruence of being a “smokescreen” when you feel like you should be “more a part of the solution” is the crux of this analysis, and we describe the resulting dissonance in more detail below.

### 3.4. Question 3: The Development of Role Identity Dissonance

We illustrate this process in Figure 1. Role identities (“I am”) provide a set of attitudes, values, beliefs, and behavioral norms that are important in fulfilling the role (“Therefore, I”). Concrete experiences in enacting the role can, in turn, either reinforce or potentially threaten the stability of the role identity [33]. Where experiences do not align with the perceived functional requirements or attributes of the role, conflict can create a sense of cognitive dissonance [11,33] that we term role identity dissonance. Role identity dissonance develops when there is a conflict between perceived functional requirements (or attributes) of the role and the subjective appraisal of the self in fulfilling these attributes. Shawn’s story at the beginning of the paper offers an illustrative example. Shawn aligns very strongly with the thrill-seeking and protecting dimensions of paramedic role identity and takes pride in his ability to “make decisions that make the difference between life and death.” His juxtaposition of the professional satisfaction from attempting to resuscitate a cardiac arrest patient (“you’re at your peak here”) with the disillusionment of attending a subsequent and much lower acuity call (“and then some guy calls for ‘I can’t sleep’”) is illustrative of the discrepancy between how he sees his role and what the role sometimes requires: “now I’m your taxi driver.” High acuity illness or injury makes up only a small proportion of a paramedic’s day-to-day caseload [34] and single instances where a specific decision or intervention is lifesaving for a patient are rarer still. This sets the stage for an “expectations versus reality” conflict in terms of what the paramedic believes their role should be and what is or is not realistically achievable. 

“Like I had never called 911 in my entire life, and to me, 911 was always like: somebody’s dead, the house is on fire. Like it’s...you would just never call 911 for the things that we see.’”(Catherine)

Another manifestation of this incongruence is in the paramedics’ ability to affect ‘upstream’ meaningful impacts on the lives of the people they encounter. For example, Nadine—who aligns very strongly with the protecting dimension—recalled a case where she attended an 18-month-old child who had died after (allegedly) being abused by her parents. Due to the fact that the family had other children in the home and the suspicion of abuse was particularly high, she spoke at length with the police and the Children’s Aid Society in the hopes of preventing similar harm to the surviving children. 

“Like I, I put a lot of time in, and it feels like it was for nothing because it’s over. It’s over. They’re not, it’s like, nobody goes—nobody has been charged. The family got their children back—their other children back. Nobody paid the price for this child’s death.”(Nadine)

Elizabeth and Shawn echoed this sentiment in explaining that calls involving children who had been deliberately harmed or killed by their caregivers as being particularly distressing: “I find those (calls) really difficult. … I see what humans do to each other, and (it’s) just appalling.” (Elizabeth). In my experience, it is uncommon for paramedics to use patients’ names in conversation, but during our interview, Shawn spoke at length about a child (referring to her by name) who had been abducted and later killed by her estranged father. 

“I found out it was her birthday right after I pronounced her. The Amber Alert was going off on all of our phones and I had to get her birthday for my paperwork, and they told me it was her birthday today.”(Shawn)

The dissonance between seeing himself as a protector and being unable to protect patients (children, in particular) began to create distress—one possible consequence of role identity dissonance that we define in Figure 1—that manifested in his home life, especially in his role as a father: “I think that’s, that’s my breaking point. I find the kid stuff is bothering me more now than it ever did before.” (Shawn).

### 3.5. Question 4: The Consequences of Role Identity Dissonance

Role identity dissonance can result in potentially significant emotional, psychological, and even existential distress. In our interviews, one common consequence of role identity dissonance was—in a word—anger. In some cases, anger was omnidirectional, stemming from a general disillusionment at the realization that paramedics often cannot affect the kind of lifesaving or otherwise meaningful impacts on patients’ lives that they had hoped for: “I went through a year of just being angry with everything” (Edward).

“I’ve become a disgruntled medic.” (Why do you see yourself that way?) “Because I hate everybody and everything. … I just feel like we’re overpaid taxi drivers. We can sometimes delay death, which, I guess is kind of cool. But ultimately, those people die, and you really didn’t do much.”(Nadiene)

“I was really disappointed once I started working and realized that the system is so abused, and I found it super upsetting because I really thought I was going to come into this job making this great positive impact on people.”(Meredith)

In other cases, the participants were angry at the training programs for—in their view—failing to prepare them for the fact that paramedic work more commonly involves responding to low acuity manifestations of complex health and social inequities. 

“The colleges teach you that you’re only going to come across problems you can fix. You have an asthmatic, here’s how to fix it. You have an anaphylactic, here’s how to fix it. … At the end of the day, I think that accounts for maybe 4% of our job?”(Jonathan)

Other participants, meanwhile, were disillusioned with the larger health and social systems and, by extension, being complicit within systems that fail the people they are supposed to help. 

“(We’re) ‘pretend help’ … It’s maybe a little bit of help, but overall, we send people back into a lot of really horrible situations.”(Elaine)

Being unable to fulfil the attributes of these roles in a professional setting sometimes meant that the participants would try to embody the roles even more strongly in other aspects of their lives.

“I’m probably more of a neurotic parent. … I freak my kids out, because I’m like ‘kay, you guys need to stay where I can see you’ and then they get freaked out, and then I’m trying to explain to them, ‘nobody is gonna hurt you, but you need to stay where I can see you.’”(Elizabeth)

Shawn echoed this sentiment, explaining that his hypervigilance for his son’s safety in everyday settings was a “red flag” that he needed to receive help for: “Somebody walked between me and my child and I was going to rip their fucking head off. Just for walking in front of my child.” Particularly for the protectors, this all-encompassing need to safeguard others in their lives had an unfortunate downstream effect. 

“My husband is not medical. He’s not, like, somebody I can go to (to talk about) work stuff. He can’t handle that. I would injure him.”(Elizabeth)

This was a common sentiment and meant that many of the participants would carry some of their most difficult experiences alone: “It’s hard to talk to people because you try so hard not to traumatize them” (Nadiene). 

### 3.6. Question 5: Reconciling Role Identity Dissonance

Our last research question examined how role identity dissonance might be reconciled. Some, such as Edward, were able to recalibrate their expectations of either the role, themselves, or both and come to a new understanding of the work.

“You know, the nursing home UTI (urinary tract infection; a routine call) is the job, whether you want it to be or not. So being angry about it is only going to affect me. You have your turn to do the ‘big calls’ and then, you know, it’s someone else’s turn.”(Edward)

Similarly, Meredith leveraged her tendency to—in her words—become “bored” with the routinized aspects of the work to take on new and interesting challenges in terms of career development: “I very quickly became bored of being a (primary care paramedic).” She went on to pursue her advanced care training and then, later, a leadership position within the service and involvement with project work. Meanwhile, other participants leaned into different role identities.

“I know that as shitty as this (job) can be, that there’s things in my life (athleticism) that I’m really good at and (that) make me happy. That won’t change.”(Sophie)

We outline the possible resolutions to role identity dissonance in Figure 1, with pathways from distress to recalibration (i.e., Edward), or from dissonance to finding fulfillment in new or other role identity (i.e., Sophie and Meredith). Still, for some participants, the consequences of a dissonant and all-encompassing paramedic role identity appeared to be irreconcilable, contributing to mental illness, disability, and lost time from work: “That’s something that came up at my last psychology session was that I do not have an identity outside of being a paramedic” (Shawn). Particularly as an insider in the community, it is painful to admit that I do not know how these “stories” end. While some participants were receiving care for the trauma they were carrying, others were not, and the long-term effects of what we have described here are largely unknown.

## 4. Discussion

Examining paramedic mental health through a role identity lens, our goals in this study were threefold: first, we sought to qualitatively explore the degree to which the definitions of paramedic role identity that have been previously described aligned with the ways in which our participants related to their work as paramedics. Second, we aimed to identify and define new potential dimensions of paramedic role identity. Finally, we sought to explore the development of role identity dissonance and its consequences for mental health and wellbeing among paramedics. We found generally very good alignment between extant role identity dimensions and the language our participants used in describing how they relate to their work as paramedics. Caregiving, thrill seeking, capacity, and, to a lesser degree, duty came through strongly in our interviews. We also identified two new potential dimensions of paramedic identity (our second objective) that describe a curiosity-driven desire to solve problems and a desire to protect vulnerable people from harm. 

Due to the fact that role identities are an important means by which we find meaning and purpose in life [15], being unable to fulfill the perceived or actual requirements, behaviors, or values (collectively, the attributes) of the role can create an existential threat to our sense of self [11]. Although circumstances varied, we saw this play out among our participants, with the commonality being the inability to realize what the participant saw as important functions or attributes of their role. We illustrate this process in Figure 1 in describing how role identity dissonance manifested and may or may not be resolved. The incongruity between an espoused role identity and what may or may not be realistically achievable in the role created cognitive dissonance that resulted in potentially significant distress, both in and out of their professional lives. For some of our participants, this cognitive dissonance and resulting distress was difficult to reconcile, while others used incongruity to reframe their expectations of themselves or the work, take on new roles within the profession, or lean into other salient role identities.

In terms of the contributions of our work, we believe that our framework extends our understanding of both identity-relevant disruptive events within role identity theory and sheds additional light on workplace stress among paramedics. Disruptive stressors within the role identity literature have tended to focus on discrete life events [20], such as a divorce or death of a spouse or transitional periods between employment and retirement [33]. Where we contribute is in offering the paramedic context as an example of how identity-relevant disruptive events can be chronic in addition to discrete. These chronic disruptive events are more nebulous, playing out in the day-to-day work of paramedics that, over the course of a career, have the potential to contribute significant emotional and psychological distress. 

Our findings also provide unique insight into why some acute and chronic workplace stressors described in the paramedic mental health literature may cause the distress that they do. For example, non-urgent calls for service have been identified as a source of workplace stress, with paramedics expressing frustration with (what they call) “system abusers” [35]. Viewed through a role identity lens, the frustration may be less with the nature of the call itself and more with the dissonance between seeing themselves as someone who responds to emergencies, protects vulnerable patients from harm, and helps people with more “legitimate” (or possibly more “fixable”) problems. At the same time, however, our analysis leaves several questions unanswered that are opportunities for further research.

First, if having a role identity is linked with a sense of meaning and purpose in life, it follows that having more (and more diverse) role identities provides greater meaning and more purpose [36]. This has been described as the role accumulation hypothesis [13], support for which has been found in population surveys examining the intersection of health and quality of life indicators with self-reported roles [37]. One important issue that is worth exploring is the potential health consequences that result from the loss of paramedic role identity (i.e., through disability) when the identity features prominently or exclusively in the person’s sense of self (i.e., “I do not have an identity outside of being a paramedic”). Although studied in the context of military service [38], an equivalent line of inquiry among paramedics has not been advanced, despite many similarities. 

Second, the relationship between role identity dissonance and mental health outcomes should be quantified. This would involve further developing the paramedic role identity scale to (1) include and psychometrically assess the new proposed dimensions of problem solver and protector and (2) develop and validating items that assess the degree to which the paramedic feels they can enact an endorsed dimension of role identity. On a practical matter, addressing the incongruence between espoused and enacted (or ‘enact-able’) role identities may lie in part at the point of entry-to-practice training. Educators can provide future paramedics with a more nuanced understanding of what is and is not realistically feasible in terms of the ability for paramedics to have (what they describe as) meaningful impacts on patients’ lives. Helping to reframe those expectations could potentially avoid future distress. At the same time, providing paramedics with specific training and resources to better manage the more prevalent low acuity manifestations of chronic health and social problems may bolster the paramedics’ self-efficacy by making them feel empowered to have different forms of meaningful impacts. 

Third, the distress we identify that results in part from role identity dissonance overlaps significantly with moral injury, which has been defined as events that involve “perpetrating, failing to prevent, bearing witness to, or learning about acts that transgress deeply held moral beliefs and expectations” [39] (p. 697). This topic has been studied extensively in military populations [40] but has not been examined in the context of paramedic work. Several of our participants described situations that could broadly be classified as morally injurious, such as cases involving children who have been deliberately harmed or killed by caregivers. The resultant moral injury is likely worsened when the paramedic conceptualizes themselves as someone who is “supposed to protect” vulnerable groups. Extending a line of inquiry that examines moral injury among paramedics within a conceptual framework of role identity is a topic worthy of further study.

## 5. Limitations

Our findings should be interpreted within the context of certain limitations. First, our work is inherently situated. We made a deliberate methodological choice to limit our investigation to a single study site, choosing depth over breadth that potentially limits the transferability of our findings. Second, qualitative research is often framed as theory-generating [41], but we made a conscious decision to adopt one conceptual framework in structuring our research questions, analysis, and the inferences we drew. Some might critique this approach as being overly rigid to the exclusion of emergent findings that are possible with a comparatively more open analytical gaze that could consider the influence of, among other things, personality, moral injury/distress, illness scripts, or other conceptual frameworks that would illuminate other aspects of health and well-being. Our findings are constrained within the scope of role identity theory, specifically. Third, in positioning our investigation under the banner of generic qualitative research, we acknowledge that we eschew the theoretical and methodological richness of, say, phenomenology, narrative inquiry, or grounded theory. As is common with qualitative inquiry, a different investigator, a different theoretical or conceptual framework, or a different methodological approach would yield different insights.

## 6. Conclusions

Amid growing concern over high rates of mental illness among paramedics, role identity theory provides a useful perspective for conceptualizing the problem. Leveraging an existing definition for paramedic role identity, we identified two new potential dimensions of role identity that appeared to resonate strongly with our participants. We also described ways in which incongruencies between an espoused and “enactable” role identity can create cognitive dissonance among paramedics. In that respect, our findings begin to shed light on why some common chronic stressors within the profession cause the distress that they do. Role identity dissonance, in turn, can result in potentially significant emotional, psychological, or existential distress—effects that can prompt an adaptive recalibration of the role or sense of self, but for some are nevertheless difficult to reconcile. Finally, in reconceptualizing identity-relevant disruptive events to include more nebulous chronic stressors, we offer a modest contribution to the theory. 

## Figures and Tables

**Figure 1 ijerph-19-02115-f001:**
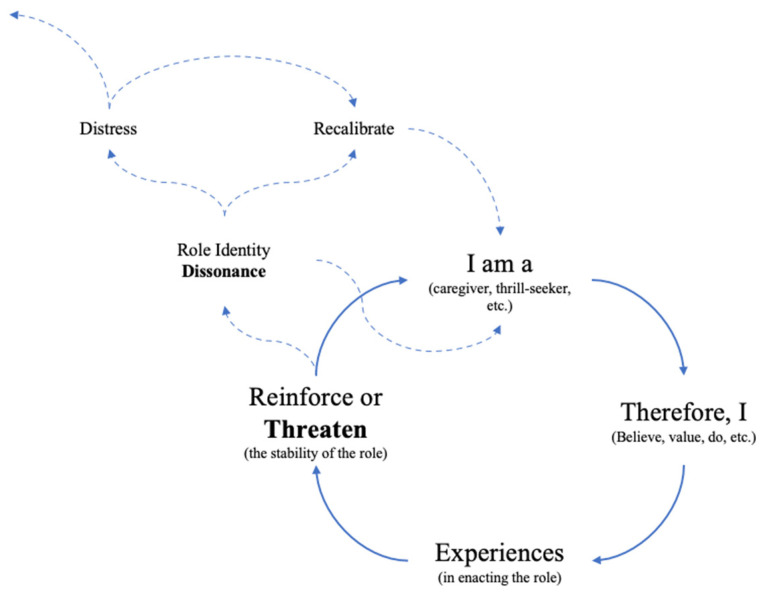
Process diagram illustrating the development and possible resolution paths of role identity dissonance in the context of paramedic role identity. “I am” refers to the concept of self in relation to an espoused role identity. The role identity, in turn, prescribes a set of beliefs, values, actions, etc. (“Therefore, I”). “Experiences” are events and interactions in enacting the role that have the potential to either reinforce or threaten the stability of the individual’s sense of self in relation to the role that they identify with. Where experiences threaten the stability of the role identity, dissonance can result and may lead to potentially significant distress or prompt a recalibration of the individual’s understanding of their sense of self, the role, or both (with potential paths indicated by dashed lines).

**Table 1 ijerph-19-02115-t001:** Demographic details for quoted participants with examples of role identity dimensions.

Pseudonym	Gender	Role Identity Dimensions	Notes
Shawn	Man	Protector, Thrill Seeking, Caregiving, Capacity: *“We do it because we want to be put on the spot (to) make a decision that makes the difference between life and death.*” “*I like the pressure. The more fucked up the call is, the calmer I get*.”	Peer Supporter, Mid-Career
Elaine	Woman	Caregiver, Protector, Capacity: “*(I’m) a stranger they can trust and rely on … (to) offer them help when they don’t think there’s any other way of getting out of whatever spot they’re in.*”	Returning from Long-Term Disability Leave, Mid-Career
Meredith	Woman	Thrill Seeking, Caregiver: “*I very quickly became bored of being a PCP … it didn’t really feel like I was helping as many people as I thought I was going to be … I need a much bigger high, like (calls) that would have excited me for a couple of days before, I’m over in, like, 10 minutes now.”*	Acting Superintendent, Mid-Career
Johnathan	Man	Thrill Seeking, Caregiving, Capacity: “*I’m resilient in the sense that 99% of this job doesn’t bother me. I’ve been in it long enough to realize you can’t fix everything, so you can’t let it bother you when it comes to calls.*” “(Being a paramedic) is important because I know that I’m one of a few in the province that can do what I do (special operations). … those skills make me really happy.”	Special Operations, Mid-Career
John	Man	Duty, Thrill Seeking, Problem Solving: “*I identified as a paramedic. It was that self sacrifice, serve the public before my needs that always came first.*” *“I’m a third generation paramedic.” “I think that’s the biggest question that drives paramedics is ‘why?’ Why is this patient the way they are now?”*	Superintendent, Peer Supporter, Late-Career
David	Man	Protecting: “*To help people, period. To be that safety net when everything else fails.”* (On becoming a supervisor): *“I came to the realization that now I’m responsible for not just the patient, myself, and my partner, but I’m responsible for all of these guys, these crews.”* (Who do you turn to after a difficult call) “*Myself. Everyone else has their own shit to deal with.”*	Superintendent, Late-Career
Elizabeth	Woman	Caregiving, Protecting, Thrill Seeking: “*Honestly, probably at the beginning, I would (have) said yeah, I’m frustrated (by non-urgent calls), but now it’s nice when you can just talk to somebody … It’s a lot of stress and pressure dealing with life and death all the time.”*	Peer Supporter, Late-Career
Edward	Man	Thrill Seeking: “*I think like anyone else, it was the expectation of, like, every call is going to be a ‘real’ call.*” *“There’s some VSAs (vital signs absent; cardiac arrest) that are almost boring ... you’re almost standing there with your hands in your pockets and you’re like ‘I am not stimulated at all’”*	Special Operations, Mid-Career
Sophie	Woman	Thrill Seeking, Capacity: “*I’ve found that I’ve had way more hot calls in (this service) than I had in (a service she worked in previously), so it was kind of resparked the job for me a little.” “I always knew that I was good at walking into a situation and controlling it, so it was kind of a cool niche, because I could do that when people are in crisis.”*	Early Career
Catherine	Woman	Thrill Seeking, Problem Solving: “*I probably thought it was a lot more dramatic than it actually is, you know, more high acuity.” “I always thought I would do something ‘sciency’”*	Acting Superintendent, Mid-Career
Nadine	Woman	Caregiving, Protecting: “*I wanted to be that person who took away the worry from people … We’re here, we’ll take care of it, you can just let it go. We’ve got this now*”	Returning from Long-Term Disability Leave, Mid-Career
Jeremiah	Man	Caregiving, Capacity, Problem Solving: *“I had worked for two or three years at the YMCA, and I’d responded to a whole bunch of medical emergencies, and I just felt a calmness about it, even with the minimal training I had. I felt like I was able to handle it.” “I’m just very curious.”*	Peer Supporter, Mid-Career
Dean	Man	Problem Solving: “*I’ve always had an interest in the science aspect. The science behind what breaks down, what works, how do you fix people, that sort of thing.” “I remember going through school and thinking’ wow, this isn’t cut and dry’, you really have to think this through and there’s a lot of judgement in it. Experience means a lot.”*	Superintendent, Late-Career
Rowan	Man	Thrill-seeking, Problem Solving: *“I’ve always liked hands on work. That’s why I like this job too; there’s a lot of skills that are hands-on. Assessing a patient is an actual skill, it’s not like I just look at somebody and know what’s wrong.”*	Mid-Career
Seamas	Man	Thrill-Seeking, Problem Solving, Protecting: *(On memorable moments) “Calls that fundamentally changed the way I practice. That my education, my background helped me figure out what was actually wrong with them. Feeling that you have an impact”* (On expectations) *“I thought I would be shot at more. I’ve been attacked a few times, but not nearly as dramatic as I would’ve hoped, but that being said, I found so much more beauty in the job. I didn’t appreciate how much thought went into paramedicine.”*	Mid-Career

Superintendent/Acting Superintendent = paramedic supervisor; Special Operations = specialized teams (e.g., tactical rescue).

## Data Availability

The data presented in this study are available upon request from the corresponding author. The data are not publicly available due to privacy restrictions and data security procedures stipulated in the Research Ethics Board (REB) review of this project.

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
