# Peer review of "Role Identity, Dissonance, and Distress among Paramedics"

_ijerph, 2022, doi:10.3390/ijerph19042115_

Round 1

Reviewer 1 Report

Dear authors,

I really enjoyed your investigation. However, the article needs a rewrite. I present some relevant issues that deserve to be analyzed.

There are many factors that influence the mental health of paramedics where a mixed study would be ideal. For example, I believe that one of the factors that greatly affect the role of paramedics is non-urgent calls. In my personal experience (as a rescuer) this was one of the factors that affected me a lot.

Qualitative analysis needs an independent judge. The results (answers to the questions) must be substantiated with literature. Results may be seriously skewed.

Attachment comments.

Best Regards

Author Response

Dear reviewer,

Thank you very much for your thoughtful and thorough review of our manuscript. We are especially grateful for your perspectives on the topic given your experience as both a psychologist and rescuer. 

You brought up a number of concerns in your review about the potential for bias in our work that could skew the interpretation and presentation of findings. Your suggestion to undertake a mixed methods study in which quantitative data and independent judges are used to render an objective interpretation of the findings is well-received, but we think the concerns that you raised are in large part attributable to differences in our philosophical approaches to research.  We agree that within a positivist or post-positivist epistemology, the objective measurement and reporting of phenomena is an important means by which to strengthen the internal and external validity of a study. Glaser and Strauss’ pioneering work on grounded theory is a notable example of using positivist thinking to lend legitimacy to early qualitative research. Where we diverge, however, is in adopting an interpretivist epistemology and its attendant axiological traditions in our current work. The distinction has important implications for the methodological choices and, we believe, explains much of the friction that may have been apparent if critiquing our work through a positivist lens. Interpretivist scholars explain that reality is experienced subjectively, negotiated collaboratively, and understood through the social construction of knowledge. Research undertaken within interpretivist framings is less concerned with the objective measurement of a verifiable truth that exists independent of the observer; instead, the subjectivity of experience and the role of the researcher as an embedded co-constructor of knowledge are embraced. The goal is not to eliminate bias, or even necessarily to substantially mitigate it, but rather to acknowledge that we all bring bias to the work that we do and to be transparent about how our preconceptions, experiences, existing knowledge, and beliefs can come to influence our interpretation of the phenomenon under study. We strive to ‘share our lens’ with the reader and engage in reflexivity practices to strengthen the credibility and trustworthiness of our findings. 

Your comments are quite helpful for us as a research team to remember that readers come from a variety of disciplinary backgrounds with different approaches to research and may find interpretivist or constructivist traditions unfamiliar. That as scientists we embrace different approaches to research is a strength of our mutual profession and contributes to greater diversity in scholarship. To aid readers who may be less familiar with the approach we used in our own work, we have added a section called Theoretical Orientations & Approach on page 3 and additional language in the researcher characteristics & reflexivitysection on page 4.

Again, and very genuinely, thank you for your deliberate and thoughtful review of our work.

Please see the attachment for further.

Justin

Reviewer 2 Report

This is an interesting paper on a timely topic. The article presents important issue of the identity, professional and social role of a paramedic. Paramedics, who play extremely important role in the emergency medical system, are often overlooked in scientific analyses, especially in the social sciences and humanities.

Social respect of the profession has been raised by media coverage of spectacular rescue operations, presenting the paramedics' activities in terms of heroism and special mission. The picture of the daily work of paramedics, can be quite different than the one shown in idealized media broadcasts: difficult working conditions, stress and professional burnout, claimant patients, physical and verbal aggression the paramedics often face, low sense of control, and at the same time unsatisfactory earnings. 

Therefore, an in-depth analysis of the mental problems of rescuers carried out from the perspective of professional identity can be extremely useful for modifying the processes of socialization of future paramedics both during education process at university and within the professional environment.

The manuscript is well prepared, with a detailed description of the research methodology and analysis of the obtained results.

Nevertheless, there are a few things that need to be improved:

1) There is a lack of detailed information on the recruiting participants. We know that there were 21 paramedics from a paramedic service recruited, who had taken an occupational stress leave; or who had been diagnosed with work-related mental health problem or were receiving treatment for it. All of them? How did the researcher contact them? Did all recruited paramedics give a consent to participate? etc.

3)  Despite the fact that there is more than one author of the manuscript, one part (2.4. Researcher Characteristics & Reflexivity) was written using 1st person singular, while others using 1st person plural. It makes confusion when reading.

4) 2.8. point (Note to Readers part) seems unnecessary, it can only be mentioned that ‘the statements of the respondents are quoted in the original’ or similar.

5) Conclusions are too general and they should refer to Results as well.

6) It could be beneficial to include more recent studies/articles and books referring to the analysed topic.

Author Response

Dear reviewer,

Thank you for your very thoughtful and constructive review of our paper. Your comments were encouraging and quite helpful in strengthening our work and we have taken care to incorporate many of the suggestions you made. In the absence of a particularly strong objection, we would prefer to keep our note to readers as a trigger warning given the sensitivity of the topic we discuss. Otherwise, please see below for more details on the edits we made in response to your very helpful suggestions. Please see the attachment for further.

Thank you again,

Justin

Reviewer 3 Report

This manuscript is well-written, and I really enjoy reading it. The authors come up with interesting idea and theory regarding a discrepancy between perceived and enacted role identity that could lead to chronic stress. The study provides detailed content that make the readers understand the paramedic sample’s frustration. However, I have some points, and would like the authors to elaborate or clarify.

  1. The theory proposed by the authors in Figure 1, which is considered a climax, seems not to be sufficiently explained. It is not clear to me, based on the authors’ assumption, what may lead the role identity dissonance to distress or recalibrate.   
  2. Still on Figure 1, why are there two “role identity dissonance” in the diagram.
  3. Where are ‘dashed’ lines mentioned in the Figure legend?
  4. As role identity is the crux of the matter in this research, other variables related to identity should be mentioned. All these sample cannot be simply treated as they have the same cohesive sense of identity before becoming paramedics. For example, they have different personalities, that would make them perceive, respond differently to the same situation. These uncontrollable variables may influence on how the authors developed the theory based on their clinical observation.

I would like the authors mention about this at least in discussion.

Author Response

Dear reviewer,

Thank you for your very thoughtful and constructive review of our work. Your comments were both kind and encouraging and the suggestions you made were very helpful in strengthening the quality of our work. We have taken your feedback to heart and incorporated many of your recommendations in our revisions. Please see the attachment for further.

Thank you again very genuinely for your review.

Justin

Round 2

Reviewer 1 Report

Dear Authors, 
I accept your arguments. You have substantiated very well my position of rejecting the article.
The changes you have made are of sufficient quality for the publication of your study.
I fully agree that research is not always objective, but as a reviewer I could not accept the article without the changes that have now been made.

Best regards,